# Scalable psychological interventions for Syrian refugees in Europe and the Middle East: STRENGTHS study protocol for a prospective individual participant data meta-analysis

Anne M de Graaff [ID],[1] Pim Cuijpers [ID],[1] Ceren Acarturk [ID],[2] Aemal Akhtar [ID],[3] Mhd Salem Alkneme [ID],[4] May Aoun,[5] Manar Awwad,[6] Ahmad Y Bawaneh,[6] Felicity L Brown [ID],[5,7] Richard Bryant [ID],[3] Sebastian Burchert [ID],[4] Kenneth Carswell [ID],[8] Annelieke Drogendijk,[9] Michelle Engels [ID],[10] Daniela C Fuhr [ID],[11] Pernille Hansen [ID],[10] Edith van 't Hof [ID],[8] Luana Giardinelli,[6] Mahmoud Hemmo [ID],[12] Jonas M Hessling,[4] Zeynep Ilkkursun [ID],[2] Mark J D Jordans [ID],[5,7] Nikolai Kiselev [ID],[12] Christine Knaevelsrud [ID],[4] Gülsah Kurt,[2] Saara Martinmäki [ID],[9] David McDaid [ID],[13] Naser Morina [ID],[12] Hadeel Naser,[6] A-La Park [ID],[13] Monique C Pfaltz [ID],[12,14] Bayard Roberts,[11] Matthis Schick,[12] Ulrich Schnyder [ID],[12] Julia Spaaij,[12] Frederik Steen,[5] Karine Taha,[5] Ersin Uygun,[15] Peter Ventevogel [ID],[16] Claire Whitney [ID],[6] Anke B Witteveen [ID],[1] Marit Sijbrandij [ID],[1] On behalf of the STRENGTHS consortium

For numbered affiliations see end of article.

**Correspondence to**
Anne M de Graaff;
a.m.de.graaff@vu.nl

## ABSTRACT

**Introduction** The World Health Organization's (WHO) scalable psychological interventions, such as Problem Management Plus (PM+) and Step-by-Step (SbS) are designed to be cost-effective non-specialist delivered interventions to reduce symptoms of common mental disorders, such as anxiety, depression and post-traumatic stress disorder (PTSD). The STRENGTHS consortium aims to evaluate the effectiveness, cost-effectiveness and implementation of the individual format of PM+ and its group version (gPM+), as well as of the digital SbS intervention among Syrian refugees in seven countries in Europe and the Middle East. This is a study protocol for a prospective individual participant data (IPD) meta-analysis to evaluate (1) overall effectiveness and cost-effectiveness and (2) treatment moderators of PM+, gPM+ and SbS with Syrian refugees.

**Methods and analysis** Five pilot randomised controlled trials (RCTs) and seven fully powered RCTs conducted within STRENGTHS will be combined into one IPD meta-analytic dataset. The RCTs include Syrian refugees of 18 years and above with elevated psychological distress (Kessler Psychological Distress Scale (K10>15)) and impaired daily functioning (WHO Disability Assessment Schedule 2.0 (WHODAS 2.0>16)). Participants are randomised into the intervention or care as usual control group, and complete follow-up assessments at 1-week, 3-month and 12-month follow-up. Primary outcomes are symptoms of depression and anxiety (25-item Hopkins Symptom Checklist). Secondary outcomes include daily functioning (WHODAS 2.0), PTSD symptoms (PTSD Checklist for DSM-5) and self-identified problems (PSYCHLOPS). We will conduct a one-stage IPD meta-analysis using linear mixed models. Quality of evidence will be assessed using the GRADE approach, and the economic evaluation approach will be assessed using the CHEC-list.

**Ethics and dissemination** Local ethical approval has been obtained for each RCT. This IPD meta-analysis does not require ethical approval. The results of this study will be published in international peer-reviewed journals.

### Strengths and limitations of this study

► An advantage of individual participant data meta-analysis concerns the acquired power to study overall treatment effect estimates, predictors and moderators.

► Selected randomised controlled trials are intentionally similar in study design, outcome measures and target population to allow for optimal pooling of data.

► The study sample will be large (estimated >1000 participants) and relatively homogeneous (ie, Syrian refugees).

► The meta-analyses will have reduced power in case individual randomised controlled trials do not meet the required sample size.

## INTRODUCTION

Since the outbreak of the civil war in Syria, over 13 million Syrians have been displaced. The leading destinations of Syrians displaced across borders include Turkey (3.6 million), Lebanon (910 600) and Jordan (654 700). One million fled to Europe by boat or overland.[1] Refugees often reside in unstable and insecure locations in urban areas or refugee camps,[2–4] and may face uncertainty about their legal status in seeking asylum.[5] In this paper, we use 'Syrian refugees' to refer to Syrians who have been granted a refugee status and to those whose request for sanctuary still has to be processed. Due to their exposure to potentially traumatic events and other stressors before, during, and after displacement, refugees are at risk of developing mental disorders.[6–9] However, health systems are often under-resourced to respond to the mental health needs of refugee populations, especially in low-income and middle-income countries, where health systems are already overburdened.[10 11] The rapidly escalated refugee crisis also poses challenges for health systems in well-resourced settings in Europe,[12] where mental health services are underused due to barriers such as communication difficulties and fear of stigma, and access is hampered by lack of culturally appropriate mental health services and long wait lists.[13–15]

The Syrian REfuGees MeNTal HealTH Care Systems (STRENGTHS) research consortium aims to evaluate the effectiveness, cost-effectiveness and implementation of scalable psychological interventions among Syrian refugees in Europe and the Middle-East.[16] The interventions have been developed by the World Health Organization (WHO) to scale up coverage for priority conditions, including depression and anxiety,[17] to improve access to mental health and psychosocial support in resource-constrained settings.[18] Scalable interventions are strongly protocolised and use evidence-based psychological techniques from therapies such as cognitive–behavioural therapy and interpersonal therapy. Typical features include the duration of the intervention (ie, few sessions instead of lengthy treatment programmes), the target of the intervention (ie, transdiagnostic instead of focused on a single disorder), and the delivery agent (ie, task-shifting to non-specialist providers).[17 19]

One of the interventions that is currently being tested by the STRENGTHS consortium is Problem Management Plus (PM+), which can be delivered by trained and supervised non-specialists to reduce psychological distress in communities affected by adversity.[20] The first randomised controlled trials (RCTs) on PM+ have been conducted in communities affected by violence in Pakistan and Kenya,[21–23] and in a disaster-prone community in Nepal.[24] PM+ for individuals led to improvements in anxiety, depression, post-traumatic stress disorder (PTSD), psychosocial functioning and self-identified problems 3 months after the intervention.[21 23] Although PM+ was more effective in reducing symptoms of common mental disorders, it was also costlier compared with CAU alone.[25] PM+ delivered in groups (gPM+) to women in Pakistan was effective in reducing symptoms of anxiety, depression, and self-identified problems, but not for symptoms of PTSD.[22] In Nepal, gPM+ for men and women was also associated with improvements in psychological distress, depression and 'heart-mind' problems, but not for psychosocial functioning and symptoms of PTSD.[24] Another non-specialist delivered intervention based on PM+ that is currently being tested by the STRENGTHS consortium is the digital Step-by-Step (SbS) intervention.[26] SbS is a minimally guided e-mental health intervention with 'e-helper' support. Pilot studies on SbS in Lebanon, including Syrian participants, found that the intervention was acceptable,[27] and likely effective in improving psychosocial outcomes.[28]

Pilot studies among Syrian refugees carried out by the STRENGTHS consortium indicated that PM+ delivered by peers is an acceptable and feasible intervention, and likely effective in reducing psychological distress.[29–32] Currently, fully-powered (ie, based on a priori power analysis) pragmatic RCTs among adult Syrian refugees in Europe and the Middle East are being conducted to test the effectiveness of PM+ in individual format in Switzerland and the Netherlands,[33] PM+ in group format (gPM+) in Jordan[34] and Turkey,[35] and the digital SbS intervention in Germany, Sweden and Egypt.[16] The interventions are compared with care as usual (CAU). CAU refers to all (mental) health services available to refugees in the setting where the pragmatic trial is conducted.[36] Although study sites differ with regard to the availability of services, barriers to accessing care have been identified across host countries.[13 15 37]

Initial results of PM+/gPM+ delivered by trained non-specialists are promising when looking at overall effects. However, individual differences in symptom severity, sociodemographic background, comorbidities, living circumstances and life events may result in individual differences in treatment response. Single trials do not have sufficient power to detect genuine interaction effects between participant characteristics and treatment outcome.[38] To determine which individuals are more or less likely to benefit from these scalable psychological interventions, an individual participant data (IPD) meta-analytic approach can be used.[39 40] This approach does not rely on aggregate (study-level) data used in conventional meta-analysis techniques, but uses the original data of each individual participant (participant-level) from eligible RCTs. The IPD meta-analysis approach offers greater statistical power and precision to better understand the predictive nature of individual characteristics on treatment outcome.[38 41] Furthermore, it also allows us to get a better estimate of overall effectiveness and cost-effectiveness, by pooling several large data sets together.

This protocol outlines a planned IPD meta-analysis on combined datasets from the STRENGTHS project, aiming to examine (1) the overall effectiveness and cost-effectiveness of scalable psychological interventions (ie, PM+, gPM+, SbS) as compared with CAU alone on improving depression and anxiety (primary outcomes),

de Graaff AM, *et al*. *BMJ Open* 2022;**12**:e058101. doi:10.1136/bmjopen-2021-058101

PTSD symptoms, psychosocial functioning and self-identified problems (secondary outcomes) among Syrian refugees and (2) sociodemographic, migratory, and clinical predictors and moderators of treatment outcome (ie, treatment-covariate interactions).

## METHODS

This protocol was pre-registered in Open Science Framework (DOI 10.17605/OSF.IO/WUHGF), and all individual RCTs were prospectively registered online (see table 1). The Preferred Reporting Items for Systematic review and Meta-Analysis Protocols (PRISMA-P) checklist[42] is appended. The IPD meta-analysis, including merging all datasets and data analyses will be conducted between January 2022 and December 2022.

### Patient and public involvement

The STRENGTHS project installed an independent external expert board (Project Advisory Board) including mental health professionals from Syria. The expert board is regularly consulted on the quality of the results and output of STRENGTHS, as well as on the dissemination and exploitation of the results. Prior to conducting the RCTs, qualitative studies among Syrian refugees in each participating country were conducted to inform intervention development, study design and recruitment strategies.

### Inclusion of datasets

We will include all STRENGTHS RCTs on the effectiveness of PM+ for individuals (Switzerland and the Netherlands), PM+ for groups (gPM+) (Jordan and Turkey), and SbS (Sweden, Germany and Egypt). These RCTs involve a relatively homogeneous population, and have similar study design and outcomes measures. The trials are presented in table 1.

### Participants and procedure

In the trials, Arabic-speaking Syrian refugees of 18 years and above are included if they report psychological distress (Kessler Psychological Distress Scale (K10 >15))[43] and impaired daily functioning (WHO Disability Assessment Schedule 2.0 (WHODAS 2.0>16)).[44] The K10 and WHODAS have been found to be valid screening instruments among refugee populations.[45–47] The fully-powered RCT on PM+ in Switzerland also includes Arabic-speaking refugees from other countries. In Jordan, an additional criterion is having a child/dependent between the ages 10–16 years.

Across trials, participants are excluded if there are signs of imminent suicide risk (PM+ manual questionnaire).[48] Furthermore, all trials except those on SbS in Sweden, Germany and Egypt exclude participants with acute medical conditions, expressed acute needs or protection risks, and indications of severe mental disorders (eg, psychotic disorders or substance-dependence) or cognitive impairment (eg, severe intellectual disability or dementia) (PM+ manual observation checklist).[48] The fully powered PM+ trial in the Netherlands also excludes participants currently receiving specialised psychological treatment. Other exclusion criteria in Switzerland include being under guardianship and inability to follow study procedures.

In line with the main aim of the STRENGTHS project, the IPD meta-analysis will include only Syrian refugees. The RCTs in Turkey, Switzerland and the Netherlands are conducted in a community setting, while the RCTs in Jordan are conducted in a refugee camp. At this moment, all pilot trials have been completed, with sample sizes ranging from 46 to 64 (total N=302). Recruitment has been completed for the fully powered trials in Turkey (N=369), Jordan (N=410) and Egypt (N±538). All trials randomise individuals 1:1 to the intervention (PM+/CAU, gPM+/CAU, or SbS/CAU) or control group (CAU only). Assessments are conducted at baseline, 1 week after the intervention, 3 months after the intervention and 12 months after baseline. The 12-month assessment was not part of the pilot RCTs. The pilot RCT in Jordan only included a baseline and 1-week follow-up assessment. Treatment conditions are further explained below.

### The interventions

The manuals for all interventions were adapted by the STRENGTHS consortium according to a framework for the cultural adaptation of psychological interventions,[49] and materials to provide training for trainers and helpers/facilitators were developed.[50 51]

### PM+ for individuals and groups

PM+ is a brief, transdiagnostic psychological intervention developed to reduce symptoms of common mental disorders such as depression, anxiety and PTSD.[20] PM+ includes five weekly sessions provided by trained non-specialist helpers. During this 5-week intervention, the participant learns skills in arousal reduction, problem-solving, behavioural activation and accessing social support. These strategies include a relaxation exercise using slow breathing (session 1), a 7-step plan to manage practical problems (session 2), behavioural activation by re-engaging with pleasant and task-oriented activities (session 3), and a strategy to strengthen social support (session 4). Homework is scheduled between sessions and strategies are reviewed in each subsequent session. Session 5 focuses on relapse prevention. The individual PM+ intervention consists of five 90 min sessions. The group PM+ intervention consists of five 2-hour sessions in groups of 6–10 participants of the same gender led by two gender-matched group facilitators.[20 51]

PM+ is delivered by Arabic-speaking peer-refugees who have completed high school, and have a background in education, social work, healthcare or another related field. In Jordan, group facilitators were members of the host community. Helpers/group facilitators in Switzerland, the Netherlands and Turkey also needed to be proficient in English or the local language (ie, German, Dutch

**Table 1** Trial descriptions

| Study site | Trial | Intervention | Setting | Inclusion criteria | Exclusion criteria | Preregistration trial | Trial dates* |
|---|---|---|---|---|---|---|---|
| Netherlands | Pilot | PM+ | Community | STRENGTHS criteria† | STRENGTHS criteria‡ | NTR: NL6665 | April 2018–March 2019 |
| | RCT | PM+ | Community | STRENGTHS criteria† | STRENGTHS criteria‡; Current receipt of specialised psychological treatment | NTR: NL7552 | February 2019–ongoing |
| Switzerland | Pilot | PM+ | Community | STRENGTHS criteria† | STRENGTHS criteria‡; Being under guardianship; Inability to follow study procedures | ClinicalTrials.gov: NCT03830008 | December 2018–March 2020 |
| | RCT | PM+ | Community | STRENGTHS criteria† + Arabic-speaking refugees 18+yrs | STRENGTHS criteria‡; Being under guardianship; Inability to follow study procedures | ClinicalTrials.gov: NCT04574466 | February 2020–ongoing |
| Turkey | Pilot | gPM+ | Community | STRENGTHS criteria† | STRENGTHS criteria‡ | ClinicalTrials.gov: NCT03567083 | September 2018–January 2019 |
| | RCT | gPM+ | Community | STRENGTHS criteria† | STRENGTHS criteria‡ | ClinicalTrials.gov: NCT03960892 | August 2019–November 2021 |
| Jordan | Pilot | gPM+ | Refugee camp | STRENGTHS criteria† + Having a child or dependent aged 10–16years | STRENGTHS criteria‡ | ANZCTR: ACTRN12619000168156 | January–April 2019 |
| | RCT | gPM+ | Refugee camp | STRENGTHS criteria† + Having a child or dependent aged 10–16years | STRENGTHS criteria‡ | ANZCTR: ACTRN12619001386123 | August 2019–July 2020 |
| Germany | Pilot | SbS | Digital | STRENGTHS criteria† | Signs of imminent suicide risk | DRKS: DRKS00017838 | December 2019–October 2020 |
| | RCT | SbS | Digital | STRENGTHS criteria† | Signs of imminent suicide risk | DRKS: DRKS00022143 | August 2020–ongoing |
| Sweden | RCT | SbS | Digital | STRENGTHS criteria† | Signs of imminent suicide risk | DRKS: DRKS00022144 | August 2020–ongoing |
| Egypt | RCT | SbS | Digital | STRENGTHS criteria† | Signs of imminent suicide risk | DRKS: DRKS00023505 | March 2021–ongoing |

*Recruitment to follow-up.
†STRENGTHS inclusion criteria = Arabic-speaking Syrian refugees 18+years + K10>15 + WHODAS 2.0>16.
‡STRENGTHS exclusion criteria = Signs of imminent suicide risk, acute medical conditions, expressed acute needs or protection risks, and/or Indications of severe mental disorders.
ANZCTR, Australian New Zealand Clinical Trials Registry; gPM+, group PM+; K10, Kessler Psychological Distress Scale; NTR, Netherlands Trial Register; pilot, pilot RCT; PM+, Problem Management Plus; RCT, fully-powered RCT; SbS, step-by-step; STRENGTHS, Syrian REfuGees MeNTal HealTH Care Systems; WHODAS 2.0, WHO Disability Assessment Schedule 2.0.

or Turkish). Helpers/group facilitators received an 8-day training of helpers (ToH) on common mental disorders, basic counselling skills, delivery of intervention strategies and self-care. The ToH was followed by practice cases. Helpers/group facilitators receive weekly supervision by trained PM+ trainers and supervisors throughout the trial. PM+ trainers/supervisors are mental healthcare professionals who received a 5-day training of trainers (ToT) on elements covered in the ToH, as well as on training and supervision skills. They receive monthly supervision by a PM+ master trainer.[20]

### Step-by-step

SbS is a five-session, digital self-help intervention for iOS, Android and Web platforms, aimed at reducing depressive symptoms. The strategies that are introduced in this 5-week intervention include psychoeducation and trying small and pleasant activities (session 1), behavioural activation (session 2), stress management (session 3), accessing social support (session 4) and positive self-verbalisation and relapse prevention (session 5). Homework practice is scheduled between sessions. WHO removed problem-solving from SbS, as it was found to be too complex to implement in a digital application with or without helper support.[26] The intervention uses an illustrated, narrative story of a fictional character that seeks help for depression from a professional. The narrative provides educative information on the strategies, supplemented by interactive exercises (eg, activity scheduling). Sessions last 30 min, and involve an introduction part, practice part, and reinforcement part with optional contact-on-demand to non-specialist, Arabic-speaking 'e-helpers' available through a text and audio messaging system. E-helpers are university students trained and supervised by mental health professionals on basic helping skills, providing basic support, and on the identification, assessment and management of risk and (serious) adverse events.

### Care as usual

The identified studies compared PM+, gPM+ or SbS in addition to CAU, to CAU alone. CAU includes all (mental) health services available to refugees across settings. In high-income countries, this includes referral to (specialised) mental health services through primary care practitioners (eg, general practitioner). In Turkey, refugees have free access to mental health services in public hospitals and migrant health centres. Refugees residing in refugee camps in Jordan can access services inside the camp provided by national and international non-governmental organisations. However, across sites, access to mental health services is hampered by numerous barriers, such as the lack of Arabic-speaking health providers in European countries, and overburdened health systems of neighbouring countries hosting most refugees.[13 14] All study participants received information about available mental health services.

### Outcomes

Similar to the individual RCTs, the primary outcomes in this IPD meta-analysis will be symptoms of depression and anxiety at 3-month follow-up, measured by the subscales of the Hopkins Symptom Checklist (HSCL-25).[52–54] Secondary outcomes include daily functioning (WHODAS 2.0), PTSD symptoms (PTSD Checklist for DSM-5; PCL-5),[55] and self-identified problems (PSYCHLOPS).[56] Measures were selected based on their acceptable validity and reliability in refugee and Arabic-speaking groups, as well as their availability in the Arabic language and prior use with Syrian refugees.[45 46 53 54 57 58] We will also examine these outcomes at the 1-week post-assessment and 12-month follow-up.

To calculate cost-effectiveness, in addition to documented resources required for intervention delivery, we will examine the pooled changes in resource use associated with use of health-related services, and time out of usual activities that have been collected using an adapted version of the Client Service Receipt Inventory (CSRI).[59] All measures were pilot tested using cognitive interviews with members of the target community.[34]

### Study-level variables

We will obtain study-level data for each trial including country and setting in which the study was conducted, and the type of intervention tested (eg, PM+, gPM+, SbS). We will thus test overall effectiveness for all interventions together as well as the differential effect of each type of intervention.

### Individual-level variables

We will extract individual-level data from each dataset to test a range of variables that may moderate the effects of the interventions on primary and secondary outcomes, to determine differential treatment effects based on participant characteristics. We will gather and synthesise all sociodemographic variables (ie, age, gender, education level, work status, marital status), clinical characteristics (ie, symptoms of depression, anxiety and PTSD, daily functioning, the exposure and type of traumatic events, post-migration stressors) and migration variables (ie, time spent in host country, refugee status).

### IPD collection and aggregation

All research teams have agreed to contribute RCT data to these IPD meta-analyses. De-identified primary datasets will be shared using secure password protected data links. Data will be stored in a secure cloud service (Surfdrive) developed for the Dutch education and research community, which can only be accessed by the IPD meta-analysis research team. Data transfer, storage and handling will follow the EU General Data Protection Regulations (GDPR).

Data accuracy of the primary datasets will be examined to explore if it matches data reported in published reports. We will specifically check frequencies of sociodemographic variables (eg, age, gender, marital status,

education level) and descriptive statistics (eg, mean scores) of continuous scales. If discrepancies between the primary dataset and published report arise, clarification will be sought from authors. After the primary datasets are checked for accuracy, all eligible datasets will be merged into the IPD meta-analytical dataset.

## Quality of evidence

We will assess the quality of evidence by applying the Grading of Recommendations, Assessment, Development and Evaluation (GRADE) approach.[60] The GRADE approach is a system for rating the quality (ie, high, moderate, low and very low) applied to a body of evidence, and not to the individual studies. GRADE offers a transparent and structured process for developing and presenting evidence summaries and for carrying out the steps involved in developing recommendations. The GRADE approach consists of rating eight criteria, including five that may lead to rating down the quality of evidence (ie, risk of bias, inconsistency, indirectness, imprecision, publication bias) and three that may lead to rating up (ie, large effect, dose–response gradient, minimal influence of residual plausible confounding). Results will be summarised in a GRADE evidence profile.[61] For the economic analysis, we will assess the economic evaluation approach using the Consensus Health Economic Criteria list (CHEC-list).[62]

## The IPD meta-analysis

All analyses will be conducted in R V.4.0.3[63] using the packages metafor[64] and lme4.[65] The economic evaluation will be conducted in STATA statistical software.[66] We will use a 'one-stage IPD meta-analytic' approach, in which the IPD of all identified studies will be merged into one large dataset. The resulting dataset is analysed in one step, as if all participants, clustered within studies, belonged to a single trial. This approach is preferred over the 'two-stage IPD meta-analytical' approach, in which aggregated summary points are pooled in an appropriate meta-analysis model. The one-stage approach allows for more advanced modelling of the moderators, can adopt more appropriate likelihood functions and has fewer assumptions.[67]

The IPD meta-analysis will be conducted according to the intention-to-treat principle (ie, all randomised participants will be included in the analyses). Missing values for outcome data will be estimated under the missing-at-random assumption, using multiple imputation (100 imputations).[68 69] To estimate the missing values, valid predictor variables such as individual clinical and sociodemographic variables will be used (eg, distress levels at baseline, age, gender). Sensitivity analyses will be conducted on complete cases only (ie, with outcome data) to test for differences between imputed and completed values, and with completers of the interventions only.

To examine the effects of PM+, gPM+ and SbS on primary and secondary outcomes at 3-month follow-up, we will perform a mixed effect linear regression model with random intercepts with each trial having a random effect and a fixed effect for the intervention and the outcome measures. The severity of depressive and anxiety symptoms, PTSD symptoms, daily functioning, and self-identified problems, and costs of care at the 3-month follow-up will be used as dependent variables, and condition (eg, treatment vs control) will be the independent variable while adjusting for baseline scores. The standardised $\beta$ coefficient indicates how many $SD$ the dependent variables (ie, anxiety, depression, PTSD, daily functioning, self-identified problems and costs of care) change per $SD$ increase in the independent variable. Thus, the larger the $\beta$ coefficient, the greater the effect of the independent variable on the respective dependent variable, with no association among the variables if $\beta$ is 0.

To ensure robustness of our findings, the analysis of the primary outcomes will be repeated using the 'two-stage IPD meta-analytic approach'. In a two-stage approach, Cohen's $d$ will be calculated for all outcomes for every single trial separately (step 1), and will then be combined to calculate the pooled effect sizes using the random-effects model (step 2).[70]

For the economic analysis incremental cost-effectiveness ratios comparing changes in effect and costs between the intervention and control group on the primary outcome will also be estimated. Uncertainty in mean differences in intervention resource use as well as the use of health and other services between baseline and 3-month follow-up between the two groups will be accounted for using bias-corrected and accelerated bootstrapping. Purchasing power parity adjusted unit costs expressed in international dollars will be applied to resource use. Discounting will not be applied given the short duration of the study. Cost-effectiveness acceptability curves will be generated to showing the likelihood that intervention is cost-effective at different willingness-to-pay thresholds.

## Participant-level and study-level moderators

We will test whether sociodemographic, clinical and migration variables moderate the effects of psychological interventions among Syrian refugees at 3-month follow-up. To examine the effects of potential moderators, we will add an interaction term between each moderator variable and condition on depressive and anxiety symptoms into the mixed-effects linear/logistic regression models. Each potential moderator variable will be added into separate bivariate models.

## Heterogeneity

We will calculate the $I^2$ statistic to assess statistical heterogeneity indicated in percentages, with 0% indicating no heterogeneity, 25% low heterogeneity, 50% moderate heterogeneity, and 75% high heterogeneity.[71] We will also calculate 95% CI around $I^2$ using the non-central chi-squared-based approach to provide the full magnitude of heterogeneity.[72]

## Small sample bias

This study protocol describes a prospective IPD meta-analysis of RCTs conducted within the STRENGTHS project. We will examine small sample bias by inspecting the funnel plot in which sample size is plotted against the effect estimates of the primary outcome measures. Egger's test of the intercept will be carried out to inspect the degree of asymmetry of the plot.[73] Lastly, we will estimate the effect size adjusted for bias using the trim and fill procedure.[74]

## Ethics and dissemination

Local ethical approval has been obtained for each RCT. This IPD meta-analysis does not require ethical approval. The results of this study will be published in international peer-reviewed journals, and will be made publicly available through the STRENGTHS website (www.STRENGTHS-project.eu).

## DISCUSSION

Scalable psychological interventions such as PM+, gPM+ and SbS for communities affected by adversity have been found to be effective in various RCTs.[21–24] Pilot studies on PM+ adapted for Syrian refugees indicated the intervention's feasibility and likely effectiveness when delivered by peer-refugees to individuals[30 31] or groups.[29 32] The STRENGTHS consortium tests the effectiveness of different types of scalable interventions (ie, PM+, gPM+ and SbS) among Syrian refugees across different settings (eg, refugee camp, community setting). The present paper describes the procedures of an IPD meta-analysis aimed at examining the overall effectiveness and cost-effectiveness of the interventions compared with CAU alone, and at examining individual participant differences in treatment response.

This study protocol has several strengths. An essential advantage of IPD meta-analytic approaches over using conventional meta-analytic approaches concerns the acquired power to study overall effect estimates, predictors and moderators, by synthesising the original data from all individual RCTs into one large dataset. Single RCTs frequently do not have enough statistical power to investigate moderators. In conventional meta-analysis, moderation analyses are commonly done using subgroup analyses of aggregate data (eg, effect sizes), but this limits statistical power and accuracy due to a loss of degrees of freedom and variability in the moderator of interest.[67] IPD meta-analysis also allows us to investigate outcome variables that primary studies have not reported. Another strength of our protocol concerns the availability of study trials with similar population, study design and outcome variables. This enables us to determine who benefits more or less from scalable psychological interventions using a meta-analytic approach that is least biased and most reliable for addressing this question.

This study has important limitations that have to be taken into account when interpreting the results. First, overall statistical power may be reduced in case individual RCTs do not meet the required sample size based on a priori power calculation. Second, study participants reside in different settings, including refugee camps with limited access to mental healthcare, and community settings in high-income countries where specialist services are widely available although not always easily accessible to refugees. Although we will register the CAU delivered at each site using the CSRI measure, it cannot be precluded that differences in CAU may affect the results. Furthermore, migratory patterns as well as the ongoing stressors might differ considerably across settings.[1 75] Third, the use of different recruitment strategies across and within trials may have affected sample composition and treatment effects.[76] Fear of stigma has been found to be one of the reasons refugees may not access mental health services.[15] For trials where participants are mainly self-selected (eg, through social media campaigns in Germany or the Netherlands), participants may also hold more positive attitudes towards mental health services. Participants in Jordan, however, were actively recruited door-to-door by members of the research team. Thus, study samples across sites may systematically differ from each other regarding their motivation to engage in mental health services, which in turn may affect the interpretation of the IPD meta-analysis results.[76] Furthermore, the results may be influenced by response-shifts at follow-up on self-report measures such as the HSCL-25 (eg, lack of unidimensionality and temporal invariance),[77] especially when response-shifts differ between treatment and control groups.[78]

This IPD meta-analysis protocol will extend our current knowledge on the effectiveness of scalable psychological interventions for refugees and other populations affect by adversity. It will allow for more accurate estimations of treatment effects, cost-effectiveness, and the exploration of important predictors and treatment moderators of scalable interventions for subgroups of refugees in Europe and the Middle East. The results of this study will be disseminated through publication in peer-reviewed journals.

**Author affiliations**
[1]Department of Clinical, Neuro- and Developmental Psychology, World Health Organization Collaborating Center for Research and Dissemination of Psychological Interventions, Amsterdam Public Health Research Institute, Vrije Universiteit Amsterdam, Amsterdam, The Netherlands
[2]Department of Psychology, Koç University, Istanbul, Turkey
[3]School of Psychology, University of New South Wales, Sydney, New South Wales, Australia
[4]Division of Clinical-Psychological Intervention, Department of Education and Psychology, Freie Universitat Berlin, Berlin, Germany
[5]Research and Development Department, War Child, Amsterdam, The Netherlands
[6]Technical Unit, International Medical Corps, London, UK
[7]Amsterdam Institute of Social Science Research, University of Amsterdam, Amsterdam, The Netherlands
[8]Department of Mental Health and Substance Abuse, World Health Organization, Geneve, Switzerland
[9]ARQ International, ARQ National Psychotrauma Centre, Amsterdam, The Netherlands

[10]International Federation of Red Cross and Red Crescent Societies Reference Centre for Psychosocial Support, Copenhagen, Denmark

[11]Department of Health Services Research and Policy, London School of Hygiene and Tropical Medicine, London, UK

[12]Department of Consultation-Liaison Psychiatry and Psychosomatic Medicine, University Hospital Zurich, Zurich, Switzerland

[13]Care Policy and Evaluation Centre, Department of Health Policy, The London School of Economics and Political Science, London, UK

[14]Department of Psychology and Social Work, Mid Sweden University, Sundsvall, Sweden

[15]Trauma and Disaster Mental Health Master Programme, Istanbul Bilgi University, Istanbul, İstanbul, Turkey

[16]Public Health, United Nations High Commissioner for Refugees, Geneva, Switzerland

**Collaborators** STRENGTHS consortium Ceren Acarturk, Aemal Akhtar, Akinçi Ahmad Bawaneh, Martha Bird, Felicity Brown, Richard Bryant, Sebastian Burchert, Pim Cuijpers, Martine van den Dool, Anne de Graaff, Annelieke Drogendijk, Daniela Fuhr, Mahmoud Hemmo, Jonas Maria Hessling, Zeynep Ilkkursun, Mark Jordans, Nikolai Kiselev, Christine Knaevelsrud, Gülşah Kurt, Saara Martinmäki, David McDaid, Cansu Mirzanlı, Trudy Mooren, Naser Morina, A-La Park, Monique Pfaltz, Bayard Roberts, Matthis Schick, Ulrich Schnyder, Marit Sijbrandij, Egbert Sondorp, Julia Spaaij, Frederik Steen, Karine Taha, Peter Ventevogel, Claire Whitney, Nana Wiedemann, Aniek Woodward.

**Contributors** AG drafted the study protocol. AG, PC, CA, AA, MSA, MAw, AB, RB, SB, ME, DF, LG, MH, JH, ZI, NK, CK, GK, DM, NM, HN, AP, MP, RB, MSch, US, JS, EU, CW, AW and MSij contributed to the original data acquisition. DM and AP provided input for the economic analysis. AG, PC, CA, AA, MSA, MAo, MAw, AB, FB, RB, SB, KC, AD, ME, DF, PH, EH, LG, MH, JH, ZI, MJ, NK, CK, GK, SM, DM, NM, HN, AP, MP, BR, MSch, US, FS, JS, KT, EU, PV, CW, AW and MSij all provided input for the study design. AG, PC, CA, AA, MSA, MAo, MAw, AB, FB, RB, SB, KC, AD, ME, DF, PH, EH, LG, MH, JH, ZI, MJ, NK, CK, GK, SM, DM, NM, HN, AP, MP, BR, MSch, US, FS, JS, KT, EU, PV, CW, AW and MSij all read, commented and approved the final protocol. MSij is the guarantor of the protocol. This protocol is written on behalf of the STRENGTHS consortium.

**Funding** The STRENGTHS project is funded under Horizon 2020—the Framework Programme for Research and Innovation (2014–2020).

**Disclaimer** The authors alone are responsible for the views expressed in this article and they do not necessarily represent the views, decisions or policies of the institutions with which they are affiliated. The European Community is not liable for any use that may be made of the information contained therein.

**Competing interests** DM: Payment to institution for advice (not related to the interventions evaluated in STRENGTHS) given to the British Association for Counselling and Psychotherapy in January 2022.

**Patient and public involvement** Patients and/or the public were involved in the design, or conduct, or reporting, or dissemination plans of this research. Refer to the Methods section for further details.

**Patient consent for publication** Not applicable.

**Provenance and peer review** Not commissioned; externally peer reviewed.

**ORCID iDs**
Anne M de Graaff http://orcid.org/0000-0001-6686-4432
Pim Cuijpers http://orcid.org/0000-0001-5497-2743
Ceren Acarturk http://orcid.org/0000-0001-7093-1554
Aemal Akhtar http://orcid.org/0000-0002-8510-3636
Mhd Salem Alkneme http://orcid.org/0000-0002-4593-6705
Felicity L Brown http://orcid.org/0000-0001-6800-1657
Richard Bryant http://orcid.org/0000-0002-9607-819X
Sebastian Burchert http://orcid.org/0000-0003-3126-5485
Kenneth Carswell http://orcid.org/0000-0001-5344-5802
Michelle Engels http://orcid.org/0000-0002-3782-1609
Daniela C Fuhr http://orcid.org/0000-0001-9020-4629
Pernille Hansen http://orcid.org/0000-0002-4782-458X
Edith van 't Hof http://orcid.org/0000-0002-5856-3573
Mahmoud Hemmo http://orcid.org/0000-0002-2958-0702
Zeynep Ilkkursun http://orcid.org/0000-0001-6691-1809
Mark J D Jordans http://orcid.org/0000-0001-5925-8039
Nikolai Kiselev http://orcid.org/0000-0003-0617-0353
Christine Knaevelsrud http://orcid.org/0000-0003-1342-7006
Saara Martinmäki http://orcid.org/0000-0003-0230-4641
David McDaid http://orcid.org/0000-0003-0744-2664
Naser Morina http://orcid.org/0000-0002-6470-4408
A-La Park http://orcid.org/0000-0002-4704-4874
Monique C Pfaltz http://orcid.org/0000-0002-7762-6226
Ulrich Schnyder http://orcid.org/0000-0003-3556-7990
Peter Ventevogel http://orcid.org/0000-0002-3567-8861
Claire Whitney http://orcid.org/0000-0002-1100-5363
Anke B Witteveen http://orcid.org/0000-0002-9636-7522
Marit Sijbrandij http://orcid.org/0000-0001-5430-9810

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
