## [Reviewer comments · BMJ Open]

ARTICLE DETAILS

TITLE (PROVISIONAL)	Scalable Psychological Interventions for Syrian Refugees in Europe and the Middle East: STRENGTHS Study Protocol for a Prospective Individual Participant Data Meta-Analysis
AUTHORS	de Graaff, Anne; Cuijpers, Pim; Acarturk, Ceren; Akhtar, Aemal; Alkneime, Mhd; Aoun, May; Awwad, Manar; Bawaneh, Ahmad; Brown, Felicity; Bryant, Richard; Burchert, Sebastian; Carswell, Kenneth; Drogendijk, Annelieke; Engels, Michelle; Fuhr, Daniela; Hansen, Pernille; van 't Hof, Edith; Giardinelli, Luana; Hemmo, Mahmoud; Hessling, Jonas; Ilkkursun, Zeynep; Jordans, Mark J. D.; Kiselev, Nikolai; Knaevelsrud, Christine; Kurt, Gulsah; Martinmaki, Saara; McDaid, David; Morina, Naser; Naser, Hadeel; Park, A-La; Pfaltz, Monique; Roberts, Bayard; Schick, Matthias; Schnyder, Ulrich; Spaaij, Julia; Steen, Frederik; Taha, Karine; Uygun, Ersin; Ventevogel, Peter; Whitney, Claire; Witteveen, Anke; Sijbrandij, Marit

VERSION 1 – REVIEW

REVIEWER	Pascal Schlechter University of Cambridge
REVIEW RETURNED	14-Nov-2021

GENERAL COMMENTS	The study protocol “Scalable Psychological Interventions for Syrian Refugees in Europe and the Middle East: STRENGTHS Study Protocol for a Prospective Individual Participant Data Meta-Analysis” outlines how the STRENGTHS research consortium aims to examine the effectiveness and cost-effectiveness of scalable WHO interventions, namely Problem Management Plus (also as group version) and Step-by-Step by conducting an individual participant data meta-analysis in Syrian refugees. The authors also specify treatment moderators that they aim to include in their meta-analysis. The manuscript is well-written, the analysis plan is clearly outlined, and evaluating mental health interventions in this vulnerable population is highly relevant. There are many strengths in this study protocol, and I only have a few points the authors may consider in a potential revision: Mental health scales developed in other populations are prone to cross-cultural measurement error when applied to refugee populations, in light of different explanation models for mental health or the use of specific idioms to describe them. Therefore, a reflection on the available psychometric evidence for the used scales appears necessary. For instance, measurement invariance was systematically tested for the Hopkins Symptom Checklist among different refugee populations (Wind et al., 2017), and evidence could be drawn from a study evaluating the Arabic version of PCL-5
---

among displaced populations in Iraq (Ibrahim et al., 2018). To understand the results thoroughly, the author should reflect on the psychometric properties on all scales that were used in the different studies, including the Kessler Psychological Distress Scale. In a similar vein, it would be beneficial to reflect on the possibility of response shifts in the follow-up assessment (e.g., Fried et al., 2016).

The authors suggest that stigma is one of the many reasons for the underutilization of mental health services among refugees. Given that some studies indicate that Syrian refugees hold more negative attitudes toward help-seeking in receiving countries, this could also influence the willingness to participate in these intervention studies. A short discussion on whether these intervention studies attracted participants who were particularly willing to take part (or whether this is not considered a problem at all) may be helpful to contextualize the results.

I was wondering whether the introduction would be clearer if it first describes the aim of STRENGTH consortium and the situation of the Syrian refugees followed by an explanation of the different interventions. While the overall situation of refugees is important, this meta-analysis will focus on Syrian refugees and learning more about their situation may be helpful for the reader. In this regard, more details about the socio-cultural background of Syrian refugees are recommended. Subheadings may also be helpful to add clarity to the introduction.

A justification for the missing at random assumption is needed (White et al., 2011). Based on this, which imputation method will be used? I appreciate the sensitivity analysis with complete cases.

Minor:

- In some fields, there are strict definition of the terms efficacy, effectiveness and efficiency. I recommend that the authors briefly introduce and justify their terminology (could be a footnote).
- It is not clear to me whether the different intervention types (PM+ gPM+, SbS) will be compared systematically or whether they will only be summarized as intervention vs. CAU.
- I recommend highlighting the follow-up time-intervals in the abstract
- Many acronyms are used making it sometimes hard to follow. The authors may find a way to reduce the complexity a bit. For instance, CBT and IPT do not need to be introduced because they have only been used once. In the abstract, SbS should be introduced because it is used but only in the introduction the reader learns that it refers to Step-by-Step.
- While care as usual is well-defined in the methods, a brief explanation would be helpful in the introduction (p. 7).
- The Hamdani et al., 2020 could be contextualized more clearly. The word "furthermore" implies that a further benefit will be listed, while the authors state that PM+ is costlier. Something like: "Although it was more effective in XYZ, it was also costlier" may add clarity.
- The term "fully-powered" should be briefly defined (i.e., what is meant? Fully-powered according to a priori power-analysis?)
- P. 8. The author state that the results are promising. They could clarify what they mean by promising.
- P. 12: Typo: Helpers/group facilitators received an (not and) eight-day training.
- I applaud the authors for the preregistration of their protocol. Will

	they follow further open science practices (e.g., providing anonymized data (if possible) or their analysis code)? Again, this is a well-written study protocol dealing with an urgently important topic that certainly addresses important gaps in the current literature, and I agree with the conclusion that “This IPD meta-analysis protocol will extend our current knowledge on the effectiveness of scalable psychological interventions for refugees.” Although the statistical procedures appear sound, I have never conducted an individual participant data meta-analysis myself and recommend consulting an independent reviewer who has practical experience with such analyses. References Fried, E. I., van Borkulo, C. D., Epskamp, S., Schoevers, R. A., Tuerlinckx, F., & Borsboom, D. (2016). Measuring depression over time... Or not? Lack of unidimensionality and longitudinal measurement invariance in four common rating scales of depression. Psychological Assessment, 28(11), 1354. Ibrahim, H., Ertl, V., Catani, C., Ismail, A. A., & Neuner, F. (2018). The validity of Posttraumatic Stress Disorder Checklist for DSM-5 (PCL-5) as screening instrument with Kurdish and Arab displaced populations living in the Kurdistan region of Iraq. BMC psychiatry, 18(1), 1-8. Wind, T. R., van der Aa, N., de la Rie, S., & Knipscheer, J. (2017). The assessment of psychopathology among traumatized refugees: measurement invariance of the Harvard Trauma Questionnaire and the Hopkins Symptom Checklist-25 across five linguistic groups. European Journal of Psychotraumatology, 8(sup2), 1321357. White, I. R., Royston, P., & Wood, A. M. (2011). Multiple imputation using chained equations: issues and guidance for practice. Statistics in Medicine, 30(4), 377-399.
--	---

REVIEWER	July Lies Monash University
REVIEW RETURNED	4-Nov-2021

GENERAL COMMENTS	 1. Page6 Line 10 – Step-by-Step (SbS) is being mentioned for the first time. 2. P9 L38: The heading “Identification of eligible studies” does not seem to match the paragraph written. 3. Content from P9 L38 (whole paragraph) and P10 L20-47 may be able to fall under the same heading. If it is appropriate, a simple table to introduce each country on their intervention, inclusion and exclusion criteria, where it is run (community vs camp), etc can be easier to read/follow. 4. P10 L20 and L43: Can you please clarify if Arabic-speaking refugees from other countries on PM+ in Switzerland and the Netherlands are excluded from the analyses? 5. P12 L24 (whole paragraph): are you able to breakdown what’s covered in each of the 5 sessions like you did for PM+ and what does it mean by “session last 30 minutes, split into two parts?” 6. P13 L6 High Income Countries (HICs) is being mentioned for the first time. 7. P13 L36: It seems primary outcomes are anxiety and depression and PTSD is a secondary outcome. Could provide further justification for primary and secondary outcomes? 8. The primary and secondary outcomes will be based on data
--

	at 3-month follow-up (P13 L31), but the pilot RCT in Jordan only included a baseline and one-week follow-up assessment (P11 L3) – how are you going to account for this statistically? 9. P15 L26 Consensus Health Economic Criteria (CHEC)-list is being mentioned for the first time. 10. With the study level variables, CAU offered may be different between HICs and LAMICs in the study. What are some protocols or inclusion/exclusion criteria to keep the CAU as consistent as possible? If it is not a possible task, maybe you can add this into the limitation. 11. I was wondering if there were Syrian researchers/co-designers/community members as part of the STRENGTHS consortium, given the sole focus on Syrian refugees. If this is the case, I was wondering about their input into the project and co-authorship and if this isn't the case encourage the authors to consider this.
--	--

VERSION 1 – AUTHOR RESPONSE

Reviewer: 1

Dr. Pascal Schlechter, University of Cambridge

Comments to the Author:

The study protocol “Scalable Psychological Interventions for Syrian Refugees in Europe and the Middle East: STRENGTHS Study Protocol for a Prospective Individual Participant Data Meta-Analysis” outlines how the STRENGTHS research consortium aims to examine the effectiveness and cost-effectiveness of scalable WHO interventions, namely Problem Management Plus (also as group version) and Step-by-Step by conducting an individual participant data meta-analysis in Syrian refugees. The authors also specify treatment moderators that they aim to include in their meta-analysis.

The manuscript is well-written, the analysis plan is clearly outlined, and evaluating mental health interventions in this vulnerable population is highly relevant. There are many strengths in this study protocol, and I only have a few points the authors may consider in a potential revision:

- **Mental health scales developed in other populations are prone to cross-cultural measurement error when applied to refugee populations, in light of different explanation models for mental health or the use of specific idioms to describe them. Therefore, a reflection on the available psychometric evidence for the used scales appears necessary. For instance, measurement invariance was systematically tested for the Hopkins Symptom Checklist among different refugee populations (Wind et al., 2017), and evidence could be drawn from a study evaluating the Arabic version of PCL-5 among displaced populations in Iraq (Ibrahim et al., 2018). To understand the results thoroughly, the author should reflect on the psychometric properties on all scales that were used in the different studies, including the Kessler Psychological Distress Scale. In a similar vein, it would be beneficial to reflect on the possibility of response shifts in the follow-up assessment (e.g., Fried et al., 2016).**

We would like to thank the reviewer for this suggestion. We have edited the text accordingly (p. 8) for the K10 and WHODAS 2.0 specifically (new text underlined):

“Participants and procedure

In the trials, Arabic-speaking Syrian refugees of 18 years and above are included if they report psychological distress (Kessler Psychological Distress Scale (K10 >15)) (Kessler et al., 2002) and impaired daily functioning (WHO Disability Assessment Schedule 2.0 (WHODAS 2.0 >16)) (WHO, 2010). The K10 and WHODAS have been found to be valid screening instruments among refugee populations (Akhtar, Cuijpers, et al., 2021; Slewa-Younan et al., 2015; Sulaiman-Hill & Thompson, 2010). The fully-powered RCT on PM+ in Switzerland also includes Arabic-speaking refugees from other countries. In Jordan, an additional criterion is having a child between the ages 10-16 years.”

We also edited the text on Outcome measures on p. 10-11 (new text underlined):

“Outcomes

In the planned IPD meta-analyses, our primary outcomes will be symptoms of depression and anxiety at 3-month follow-up, measured by the subscales of the Hopkins Symptom Checklist (HSCL-25) (Derogatis et al., 1974; Mahfoud et al., 2013; Selmo et al., 2016). Secondary outcomes include daily functioning (WHODAS 2.0), PTSD symptoms (PTSD Checklist for DSM-5; PCL-5) (Blevins et al., 2015), and self-identified problems (PSYCHLOPS) (Ashworth et al., 2004). Measures were selected based on their acceptable validity and reliability in refugee and Arabic-speaking groups, as well as their availability in the Arabic language and prior use with Syrian refugees (Acarturk et al., 2016; Akhtar, Cuijpers, et al., 2021; Ibrahim et al., 2018; Mahfoud et al., 2013; Selmo et al., 2016; Sulaiman-Hill & Thompson, 2010). We will also examine these outcomes at post-assessment and 12-month follow-up.

To calculate cost-effectiveness, in addition to documented resources required for intervention delivery, we will examine the pooled changes in resource use associated with use of health-related services, and time out of usual activities that have been collected using an adapted version of the Client Service Receipt Inventory (CSRI) (Beecham & Knapp, 1992). All measures were pilot tested using cognitive interviews with members of the target community (see de Graaff et al., 2020).”

We have also added a reflection on the validity of treatment effects based on self-report questionnaires in the discussion section (p. 15):

“Furthermore, the results may be influenced by response-shifts at follow-up on self-report measures such as the HSCL-25 (e.g., lack of unidimensionality and temporal invariance) (Fried et al., 2016), especially when response-shifts differ between treatment and control groups (e.g., Fokkema et al., 2013).”

- **The authors suggest that stigma is one of the many reasons for the underutilization of mental health services among refugees. Given that some studies indicate that Syrian refugees hold more negative attitudes toward help-seeking in receiving countries, this could also influence the willingness to participate in these intervention studies. A short discussion on whether these intervention studies attracted participants who were particularly willing to take part (or whether this is not considered a problem at all) may be helpful to contextualize the results.**

We would like to thank the reviewer for this suggestion. We have incorporated a reflection on this in the discussion section on p. 14-15 (new text underlined):

“This study has important limitations that have to be taken into account when interpreting the results. (...) Third, the use of different recruitment strategies across and within trials may have affected sample composition and treatment effects (Winhusen et al., 2012). Fear of stigma has been found to be one of the reasons refugees may not access mental health services (Due et al., 2020). For trials where participants are mainly self-selected (e.g., through social media campaigns in Germany or the Netherlands), participants may also hold more positive attitudes towards mental health services. Participants in Jordan, however, were actively recruited door-to-door by members of the research team. Thus, study samples across sites may systematically differ from each other regarding their motivation to engage in mental health services, which in turn may affect the interpretation of the IPD meta-analysis results (Winhusen et al., 2012).”

- **I was wondering whether the introduction would be clearer if it first describes the aim of STRENGTH consortium and the situation of the Syrian refugees followed by an explanation of the different interventions. While the overall situation of refugees is important, this meta-analysis will focus on Syrian refugees and learning more about their situation may be helpful for the reader. In this regard, more details about the socio-cultural background of Syrian refugees are recommended. Subheadings may also be helpful to add**

clarity to the introduction.

We agree with the reviewer that a focus on Syrian refugees at the introduction of the paper would be helpful for the reader. We have reorganized the text (see p. 5-7).

- **A justification for the missing at random assumption is needed (White et al., 2011). Based on this, which imputation method will be used? I appreciate the sensitivity analysis with complete cases.**

We added the reference of White et al., 2011 to refer to the strategy that we follow to handle missing data (see p. 12):

“Missing values for outcome data will be estimated under the missing-at-random assumption, using multiple imputation (100 imputations) (Bell et al., 2014; White et al., 2011). To estimate the missing values, valid predictor variables such as individual clinical and socio-demographic variables will be used (e.g., distress levels at baseline, age, gender). Sensitivity analyses will be conducted on complete cases only (i.e., with outcome data) to test for differences between imputed and completed values, and with completers of the interventions only.”

Minor:

- **In some fields, there are strict definition of the terms efficacy, effectiveness and efficiency. I recommend that the authors briefly introduce and justify their terminology (could be a footnote).**

We added that the STRENGTHS trials were *pragmatic* randomized controlled trials, to clarify that we evaluate the *effectiveness* of the interventions in comparison to usual care (see p. 7):

“Currently, fully-powered (i.e., based on a priori power analysis) pragmatic RCTs are being conducted among adult Syrian refugees in seven countries in Europe and the Middle East to test the effectiveness of PM+ in individual format in Switzerland and the Netherlands (de Graaff et al., 2020), PM+ in group format (gPM+) in Jordan (Akhtar et al., 2020) and Turkey (Uygun et al., 2020), and the digital intervention SbS in Germany, Sweden and Egypt (Sijbrandij et al., 2017).”

- **It is not clear to me whether the different intervention types (PM+ gPM+, SbS) will be compared systematically or whether they will only be summarized as intervention vs. CAU.**

We thank the reviewer for pointing out that this was not clear. We will evaluate overall effectiveness of all interventions together (as ‘scalable psychological interventions’) and we will also examine intervention type as study-level moderator (see p. 11, new text underlined):

“Study-level variables

We will obtain study-level data for each trial including country and setting in which the study was conducted, and the type of intervention tested (e.g., PM+, gPM+, SbS). We will thus test overall effectiveness for all interventions together as well as the differential effect of each type of intervention.”

- **I recommend highlighting the follow-up time-intervals in the abstract**

We agree with the reviewer that this information in the abstract is important. We have edited the text as follows (new text underlined):

“Participants are randomized into the intervention or care as usual control group, and complete follow-up assessments at 1-week, 3-month and 12-month follow-up.”

- **Many acronyms are used making it sometimes hard to follow. The authors may find a way to reduce the complexity a bit. For instance, CBT and IPT do not need to be introduced because they have only been used once. In the abstract, SbS should be introduced because it is used but only in the introduction the reader learns that it refers to Step-by-Step.**

We thank the reviewer for pointing this out. We have reduced the number of acronyms (e.g., CBT, IPT, LAMIC, HIC, MHPSS) and have introduced Step-by-Step in the abstract.

- **While care as usual is well-defined in the methods, a brief explanation would be helpful in the introduction (p. 7).**

Following the reviewer's advice, we have added a brief description of care as usual in the introduction on p. 7 (relevant text underlined):

“Currently, fully-powered (i.e., based on a priori power analysis) pragmatic RCTs are being conducted among adult Syrian refugees in seven countries in Europe and the Middle East to test the effectiveness of PM+ in individual format in Switzerland and the Netherlands (de Graaff et al., 2020), PM+ in group format (gPM+) in Jordan (Akhtar et al., 2020) and Turkey (Uygun et al., 2020), and the digital Step-by-Step (SbS) intervention in Germany, Sweden and Egypt (Sijbrandij et al., 2017) in comparison to care as usual (CAU). CAU refers to all (mental) health services available to refugees in the setting where the pragmatic trial is conducted (Cuijpers et al., 2021). Although study sites differ with regard to the availability of services, barriers to accessing care have been identified across host countries (Cratsley et al., 2021; Due et al., 2020; Satinsky et al., 2019).”

- **The Hamdani et al., 2020 could be contextualized more clearly. The word “furthermore” implies that a further benefit will be listed, while the authors state that PM+ is costlier. Something like: “Although it was more effective in XYZ, it was also costlier” may add clarity.**

We would like to thank the review for his suggestion. We have edited the text on p. 6 as follows (new text underlined):

“Although PM+ was more effective in reducing symptoms of common mental disorders, it was also costlier compared to CAU (Hamdani et al., 2020).”

- **The term “fully-powered “should be briefly defined (i.e., what is meant? Fully-powered according to a priori power-analysis?)**

We have clarified this by adding the following text (new text underlined), see p. 7:

“Currently, fully-powered (i.e., based on a priori power analysis) pragmatic RCTs are being conducted among adult Syrian refugees in seven countries in Europe and the Middle East to test the effectiveness of PM+ in individual format in Switzerland and the Netherlands (de Graaff et al., 2020), PM+ in group format (gPM+) in Jordan (Akhtar et al., 2020) and Turkey (Uygun et al., 2020), and the digital intervention SbS in Germany, Sweden and Egypt (Sijbrandij et al., 2017).”

- **P. 8. The author state that the results are promising. They could clarify what they mean by promising.**

We have edited the sentence as follows (p. 7):

“Initial results of PM+ delivered by trained peer-refugees to individuals and groups are promising when looking at overall effects.”

- **P. 12: Typo: Helpers/group facilitators received an (not and) eight-day training.**

We would like to thank the reviewer for flagging this. We have deleted -d.

- **I applaud the authors for the preregistration of their protocol. Will they follow further open science practices (e.g., providing anonymized data (if possible) or their analysis code)?**

We have added the following section on p. 16:

“Data

The Vrije Universiteit Amsterdam (VU) will keep a central data repository of all data collected in the STRENGTHS project. The data will be available upon reasonable request to the STRENGTHS consortium. Data access might not be granted to third parties when this would interfere with relevant data protection and legislation in the countries participating in this project and any applicable EU legislation regarding data protection. Interested researchers can contact Dr Marit Sijbrandij at e.m.sijbrandij@vu.nl to initiate the process.”

Again, this is a well-written study protocol dealing with an urgently important topic that certainly addresses important gaps in the current literature, and I agree with the conclusion that “This IPD meta-analysis protocol will extend our current knowledge on the effectiveness of scalable psychological interventions for refugees.” Although the statistical procedures appear sound, I have never conducted an individual participant data meta-analysis myself and recommend consulting an independent reviewer who has practical experience with such analyses.

References

Fried, E. I., van Borkulo, C. D., Epskamp, S., Schoevers, R. A., Tuerlinckx, F., & Borsboom, D. (2016). Measuring depression over time... Or not? Lack of unidimensionality and longitudinal measurement invariance in four common rating scales of depression. *Psychological Assessment*, 28(11), 1354.

Ibrahim, H., Ertl, V., Catani, C., Ismail, A. A., & Neuner, F. (2018). The validity of Posttraumatic Stress Disorder Checklist for DSM-5 (PCL-5) as screening instrument with Kurdish and Arab displaced populations living in the Kurdistan region of Iraq. *BMC psychiatry*, 18(1), 1-8.

Wind, T. R., van der Aa, N., de la Rie, S., & Knipscheer, J. (2017). The assessment of psychopathology among traumatized refugees: measurement invariance of the Harvard Trauma Questionnaire and the Hopkins Symptom Checklist-25 across five linguistic groups. *European Journal of Psychotraumatology*, 8(sup2), 1321357.

White, I. R., Royston, P., & Wood, A. M. (2011). Multiple imputation using chained equations: issues and guidance for practice. *Statistics in Medicine*, 30(4), 377-399.

Reviewer: 2

Dr. July Lies, Monash University

Comments to the Author:

Thank you for inviting me to review your study. This paper is compelling and a pleasure to read because it has clear rationale, coherent design, and extensive pilot work. The STRENGTH study seems simple but it's not; it's elegant and powerful.

Page6 Line 10 – Step-by-Step (SbS) is being mentioned for the first time.

We thank the reviewer for pointing this out. We have added Step-by-Step in full (see p. 6).

P9 L38: The heading “Identification of eligible studies” does not seem to match the paragraph written.

We agree with the reviewer that this heading is not accurate for our prospective IPD meta-analysis. We have changed it to (see p. 8) “Inclusion of datasets”.

Content from P9 L38 (whole paragraph) and P10 L20-47 may be able to fall under the same heading. If it is appropriate, a simple table to introduce each country on their intervention, inclusion and exclusion criteria, where it is run (community vs camp), etc can be easier to read/follow.

We agree with the reviewer and have added a table (Table 1) to provide details for each trial. See also our response to the third point in this letter.

P10 L20 and L43: Can you please clarify if Arabic-speaking refugees from other countries on PM+ in Switzerland and the Netherlands are excluded from the analyses?

The focus of this IPD meta-analysis is the effectiveness of scalable psychological interventions among Syrian refugees, which is in line with the aims of the STRENGTHS project. Therefore, other Arabic-speaking refugees that may be included in the Swiss main RCT will not be included in our IPD meta-analysis. We have edited our manuscript as follows (p. 9, new text underlined):

“In line with the main aim of the STRENGTHS project, the IPD meta-analysis will include only Syrian refugees.”

P12 L24 (whole paragraph): are you able to breakdown what’s covered in each of the 5 sessions like you did for PM+ and what does it mean by “session last 30 minutes, split into two parts?”

We have included a more detailed description on the 5 SbS sessions on p. 10:

“The strategies that are introduced in this 5-week intervention include psychoeducation and trying small and pleasant activities (session 1), behavioural activation (session 2), stress management (session 3), accessing social support (session 4), and positive self-verbalization and relapse prevention (session 5). Homework practice is scheduled between sessions.”

We have also explained the different parts in the sessions (new text underlined, p. 10):

“Sessions last 30 minutes, and involve an introduction part, practice part, and reinforcement part with optional contact-on-demand to non-specialist, Arabic-speaking “e-helpers” available through a text and audio messaging system.”

P13 L6 High Income Countries (HICs) is being mentioned for the first time.

We thank the reviewer for pointing this out. We have now expanded ‘high income countries’, and deleted the abbreviation.

P13 L36: It seems primary outcomes are anxiety and depression and PTSD is a secondary outcome. Could provide further justification for primary and secondary outcomes?

As described in our introduction section, WHO’s scalable psychological interventions are transdiagnostic, aiming at ‘psychological distress’ individuals may experience after adversity. The primary outcomes depression and anxiety are in line with previous RCTs on PM+ (Bryant et al., 2017; Rahman et al., 2016, 2019), as well as the individual RCTs conducted in STRENGTHS (see p. 11, new text underlined):

“Similar to the individual RCTs, the primary outcomes in this IPD meta-analysis will be symptoms of depression and anxiety at 3-month follow-up, measured by the subscales of the Hopkins Symptom Checklist (HSCL-25) (Derogatis et al., 1974; Mahfoud et al., 2013; Selmo 2016).”

The primary and secondary outcomes will be based on data at 3-month follow-up (P13 L31), but the pilot RCT in Jordan only included a baseline and one-week follow-up assessment (P11 L3) – how are you going to account for this statistically?

We will include the pilot RCT of Jordan with only pre-post assessments. Although this trial will not contribute to the main endpoint (3-month follow-up), it will be included in the analysis for the 1-week post-assessment. We added the following sentence to the Outcome section on p. 11:

“We will also examine these outcomes at the 1-week post-assessment and 12-month follow-up.”

P15 L26 Consensus Health Economic Criteria (CHEC)-list is being mentioned for the first time.

We thank the reviewer for pointing this out. We have expanded the abbreviation of the CHEC-list on p. 12.

With the study level variables, CAU offered may be different between HICs and LAMICs in the study. What are some protocols or inclusion/exclusion criteria to keep the CAU as consistent as possible? If it is not a possible task, maybe you can add this into the limitation.

We agree with the reviewer that CAU may be different between HICs and LAMICs. We have now reflected on this in several sections in our manuscript:

Introduction p. 7, including a reference to a study that investigates the impact of using different types of CAU in RCTs:

“The interventions are compared with care as usual (CAU). CAU refers to all (mental) health services available to refugees in the setting where the pragmatic trial is conducted (Cuijpers et al., 2021). Although study sites differ with regard to the availability of services, barriers to accessing care have been identified across host countries (Cratsley et al., 2021; Due et al., 2020; Satinsky et al., 2019).”

All RCTs assessed health care use at all assessment time points using the Client Service Receipt Inventory, so we will be able to describe the use of mental health care across trials and conditions (see p. 11). We have also provided the different types of CAU across study settings as a limitation in our discussion on p. 15:

“Second, study participants reside in different settings, including refugee camps with limited access to mental health care, and community settings in high-income countries where specialist services are widely available although not always easily accessible to refugees. Although we will register the CAU delivered at each site using the CSRI measure, it cannot be precluded that differences in CAU may affect the results. Furthermore, migratory patterns as well as the ongoing stressors might differ considerably across settings.”

I was wondering if there were Syrian researchers/co-designers/community members as part of the STRENGTHS consortium, given the sole focus on Syrian refugees. If this is the case, I was wondering about their input into the project and co-authorship and if this isn't the case encourage the authors to consider this.

We have added a section on “Patient and public involvement” on p. 8. We would also like to refer to our response to comment 5 in this letter, where we state that we included individuals from the target community in the development of the research, including a Project Advisory Board with mental health professionals from Syria. We would also like to confirm that we included (and will include) co-authors from Syria in this manuscript and in publications of the results of this IPD meta-analysis.

We have made some minor additional changes on the following pages:

- We edited our limitations section (p. 14-15, new text underlined):
 “Overall statistical power may be reduced in case individual RCTs do not meet the required sample size based on a priori power calculation.”
- We edited some of the e-mail addresses of co-authors.
- We have added the outcome measures to the abstract: “Primary outcomes are symptoms of depression and anxiety (25-item Hopkins Symptom Checklist). Secondary outcomes include daily functioning (WHODAS 2.0), PTSD symptoms (PTSD Checklist for DSM-5), and self-identified problems (PSYCHLOPS).”

VERSION 2 – REVIEW

REVIEWER	Pascal Schlechter University of Cambridge
REVIEW RETURNED	27-Jan-2022

GENERAL COMMENTS	The authors have done an excellent job incorporating the recommendations of the editor and reviewers in their revision. I applauded the authors for this important contribution to the literature. That being said, I cannot unequivocally conclude whether all proposed procedures within the IPD meta-analysis framework are sound because I lack expertise in this field. For example, the authors state that all participants are treated as clustered within studies. As far as I know, inference in random-effects models requires a large number of studies in random-effect meta-analyses. I do not know whether one-stage IPD meta-analyses are better able to handle small numbers of studies or whether certain estimators may be able to account for this. Surely, the authors will conduct a range of sensitivity analyses so that they may have an indication of bias introduced by any chosen procedure. Minor: The abbreviation SbS can be introduced in the first sentence of the abstract where Step-by-Step is mentioned first. Thanks for the opportunity to review this relevant manuscript.
---

VERSION 2 – AUTHOR RESPONSE

Reviewer: 1

Dr. Pascal Schlechter, University of Cambridge

Comments to the Author:

The authors have done an excellent job incorporating the recommendations of the editor and reviewers in their revision. I applauded the authors for this important contribution to the literature.

That being said, I cannot unequivocally conclude whether all proposed procedures within the IPD meta-analysis framework are sound because I lack expertise in this field. For example, the authors state that all participants are treated as clustered within studies. As far as I know, inference in random-effects models requires a large number of studies in random-effect meta-analyses. I do not know whether one-stage IPD meta-analyses are better able to handle small numbers of studies or whether certain estimators may be able to account for this. Surely, the authors will conduct a range of sensitivity analyses so that they may have an indication of bias introduced by any chosen procedure.

We would like to thank the reviewer for their comments. We would like to confirm that we will use a one-stage IPD approach as the preferred method in IPD meta-analysis, but will also conduct a two-

stage IPD meta-analysis to ensure robustness of our findings (see p. 12) (Burke et al., 2017; Riley et al., 2010).

Burke, D. L., Ensor, J., & Riley, R. D. (2017). Meta-analysis using individual participant data: one-stage and two-stage approaches, and why they may differ. *Statistics in Medicine*, 36(5), 855–875. <https://doi.org/10.1002/sim.7141>

Riley, R. D., Lambert, P. C., & Abo-Zaid, G. (2010). Meta-analysis of individual participant data: Rationale, conduct, and reporting. *BMJ*, 340(7745), 521–525. <https://doi.org/10.1136/bmj.c221>

Minor: The abbreviation SbS can be introduced in the first sentence of the abstract where Step-by-Step is mentioned first.

Thank you for spotting the abbreviation of SbS. We have adjusted that in the abstract.

Thanks for the opportunity to review this relevant manuscript.

We would also like to notify the editor that the study period to merge the data and conduct the analyses has been changed from January 2022-July 2022, to January 2022-December 2022 (see p. 7), to allow inclusion of all the 12-month follow-up assessments.